# Daily Yogurt Consumption Improves Glucose Metabolism and Insulin Sensitivity in Young Nondiabetic Japanese Subjects with Type-2 Diabetes Risk Alleles

**DOI:** 10.3390/nu10121834

**Published:** 2018-11-29

**Authors:** Daiki Watanabe, Sachi Kuranuki, Akiko Sunto, Naoki Matsumoto, Teiji Nakamura

**Affiliations:** 1Department of Nutrition and Dietetics, Kanagawa University of Human Service, Kanagawa 238-8522, Japan; d2watanabe@marianna-u.ac.jp (D.W.); suntou-a@kuhs.ac.jp (A.S.); nakamura-t@kuhs.ac.jp (T.N.); 2Department of Pharmacology, St. Marianna University School of Medicine, Kanagawa 216-8511, Japan; matsumoto@marianna-u.ac.jp

**Keywords:** type 2 diabetes susceptibility gene, genetic risk score, yogurt, postprandial plasma glucose, insulin sensitivity

## Abstract

This study investigated whether the association between postprandial plasma glucose (PPG) is affected by five type 2 diabetes mellitus (T2DM) susceptibility genes, and whether four weeks of yogurt consumption would affect these responses. We performed a single-arm intervention study in young nondiabetic Japanese participants, who consumed 150 g yogurt daily for four weeks, after which a rice test meal containing 50 g carbohydrate was administered. PPG and postprandial serum insulin (PSI) were measured between 0 and 120 mins at baseline and after the intervention. Genetic risk was evaluated by weighted genetic risk score (GRS) according to published methodology, and participants were assigned to one of two groups (*n* = 17: L-GRS group and *n* = 15: H-GRS group) according to the median of weighted GRS. At baseline, the H-GRS group had higher glucose area under the curve_0–120 min_ after intake of the test meal than the L-GRS group (2175 ± 248 mg/dL.min vs. 1348 ± 199 mg/dL.min, *p* < 0.001), but there were no significant differences after the yogurt intervention. However, there was an improvement in PSI in the H-GRS group compared with baseline. These results suggest that habitual yogurt consumption may improve glucose and insulin responses in nondiabetic subjects who have genetically higher PPG.

## 1. Introduction

Rice is staple food and the main contributor to high carbohydrate and energy intake in many countries especially in Asia [1]. It was reported that East Asians have high insulin sensitivity, however have a limited innate capacity of insulin secretion [2]. 

Some epidemiologic studies have shown that the intake of dairy products, such as milk and yogurt, may protect against diabetes and the insulin resistance syndrome [3,4]. In addition, recent studies have suggested that gut bacteria play a fundamental role in diseases such as obesity and diabetes [5], but these studies findings are still inconsistent with regard to the importance of this for diabetes in humans [6].

Type 2 diabetes mellitus (T2DM) is a multifactorial disease arising from the complex interaction between an individual’s genetic makeup and environmental factors [7,8]. Numerous genome-wide association studies (GWAS) of T2DM have been conducted worldwide, and over 80 susceptibility loci have been identified [9]. More recently, GWAS-derived loci have been found to be associated with T2DM in Japanese population [10]. Interestingly, most of the diabetes risk genes affect β-cell function, and this supports the hypothesis that the principal effect of genetics in the development of T2DM is to impair insulin secretion [11]. Typically, each copy of a susceptibility allele at one of these loci is associated with 15–20% increase in the risk of T2DM [12]. The genetic risk of T2DM has been evaluated using the genetic risk score (GRS) [13], with a high GRS for T2DM being associated with a greater incidence of diabetes [14].

Several studies have investigated the interaction between genetic and environmental risk factors in the Western population [15,16]. These have demonstrated a greater area under the curve (AUC) for glucose following a meal in T allele carriers of the transcription factor 7-like 2 (TCF7L2) gene rs7903146 variant [17]. However, other studies have evaluated the impact of T2DM susceptibility genes on glucose and insulin responses following an oral glucose tolerance test (OGTT) [18,19]. Finally, observational studies have examined the association between T2DM susceptibility gene variants and the intake of macronutrients, such as dietary carbohydrates and fiber, in Western population [20]. However, to our knowledge, limited data have been obtained regarding potential gene-diet interactions in the Japanese population.

We aimed to investigate whether five validated T2DM susceptibility genes (KCNQ1 [potassium voltage-gated channel, KQT-like subfamily, member 1] rs2237892, CDKAL1 [CDK5 regulatory subunit associated protein 1-like 1] rs2206734, CDKN2B [cyclin-dependent kinase inhibitor 2B] rs2383208, UBE2E2 [ubiquitin-conjugating enzyme E2] rs6780569, and IGF2BP2 [insulin-like growth factor 2 mRNA binding protein 2] rs1470579) variants are associated with glucose and insulin responses following rice intake in young nondiabetic Japanese subjects, and whether four weeks of yogurt consumption would affect these responses.

## 2. Materials and Methods

### 2.1. Study Participants

In order to be eligible for enrollment, participants had to be 20–29 years old and have a normal glucose tolerance. Study enrollment took place from March 14, 2014, to May 1, 2014. Participants were included according to judgmental sampling and excluded if they had a food allergy to dairy products or rice, were taking medicine, were smokers, had abnormal glucose tolerance during the previous year, had a body mass index (BMI) >30 kg/m^2^, or habitually consumed dairy products (≥200 mL milk or 100 g yogurt per week).

The study protocol was approved by the Ethics Committee of St. Marianna University School of Medicine (No.2654) and Kanagawa University of Human Services (No.25-038). In addition, the trial was registered with the University Hospital Medical Information Network in Japan Clinical Trial Registry (UMIN000021331). The following are available online at, https://upload.umin.ac.jp/cgi-open-bin/ctr_e/ctr_view.cgi?recptno=R000024599. All study participants gave written informed consent before the initiation of the study.

### 2.2. Study Design

This study consisted of three phases (Figure 1). In the first phase, the participants underwent a control meal test as the baseline. In the second step, the participants consumed a 150 g yogurt, containing 72 kcal energy, 6.8 g protein, 10.2 g carbohydrate, and 0 g fat, daily for four weeks, as the test intervention. The participants were asked not to change their personal behavior during the second phase. In the final phase, the control meal test was carried out again. The biological effects of consuming yogurt for four weeks were then evaluated.

### 2.3. Study Procedure

Participants ate their evening meal between 19:00 and 21:00 on the day before the control meal test, after which they were only permitted water. The following morning, after a 12-h overnight fast, participants ingested the control meal (147 g rice), which contained 217 kcal energy, 50 g carbohydrate, and 150 mL water.

Blood samples were drawn for measurement of high-density lipoprotein cholesterol (HDL-C), low-density lipoprotein cholesterol (LDL-C), total cholesterol (TC), insulin, glucose, hemoglobin A1c (HbA1c), and triglyceride (TG) after the overnight fast. Postprandial plasma glucose (PPG) and serum insulin were measured after ingestion of the control meal. Blood samples were analyzed by a central laboratory (SRL Inc., Tokyo, Japan) between May 1, 2014, and November 3, 2014.

### 2.4. DNA Extraction and Genotyping

DNA samples were obtained from the oral mucosa using a swab and DNA was extracted using a QIAamp DNA Mini Kit (Qiagen, Hilden, Germany) according to the manufacturer’s instructions. In this study, real-time polymerase chain reaction (PCR) and melting curve analyses were performed in a closed tube on a LightCycler 3.5 (Roche Diagnostics, Penzberg, Germany). The presence of alleles for five T2DM susceptibility genes was identified by HybProbe assay. The conditions for thermal cycling were as follows: initial denaturation for 10 min (to activate the fast-start Taq polymerase), followed by amplification, which included a denaturation step at 95 °C for 10 s, an annealing step at 56 °C for 15 s, and extension at 72 °C for 10 s. The amplification was followed by the construction of a melting curve, which started at 40 °C for 30 s, and cooling at 40 °C for 30 s. The primer and probe list were shown in Appendix A.

Yamauchi et al. reported that seven types of GWAS-derived loci were associated with T2DM in a Japanese population [10]. For this study, we excluded the rare variants, such as TCF7L2, because our study would not be of sufficient size to detect these [21]. Therefore, we included five T2DM susceptibility genes (KCNQ1 rs2237892, CDKAL1 rs2206734, CDKN2B rs2383208, UBE2E2 rs6780569, and IGF2BP2 rs1470579).

To evaluate the genetic risk associated with these five T2DM susceptibility genes, GRS was calculated according to published methodology [13,14]. Each single nucleotide polymorphism was recoded as 0, 1, or 2, according to the number of risk alleles for diabetes present, and the GRS thus ranges from 0–10. Moreover, we were calculated weighted GRS as weighted sum of the number of risk alleles possessed by an individual, in which the weight was taken as the natural log of the odds ratio (OR) of diabetes incidence associated with each individual risk alleles [22]. We used published ORs from previous study in Japanese population to calculate weighted GRS [10].

### 2.5. Study Outcomes

Primary and secondary outcomes were defined as glucose AUC and homeostasis model assessment of insulin resistance (HOMA-IR) over the four weeks period of yogurt consumption, respectively. The AUC were calculated by the trapezoidal rule. Insulin sensitivity was estimated by two methods: the reciprocal of the fasting insulin concentration, and the HOMA, calculated using the following formulas: HOMA-IR = fasting plasma glucose (FPG) [mg/dL] × fasting serum insulin (FSI) [μg/mL]/(18 × 22.5); homeostasis model assessment of beta cell function (HOMA-β) = 360 × FSI [μg/mL]/(FPG [mg/dL] − 63) [23]. The Insulinogenic Index (IGI) was calculated as the ratio of the change in insulin to the change in glucose between 0 and 30 min (Δ_I0-30_/Δ_G0-30_). The incremental AUC insulin/glucose (IncAUC_ins/glu_) response was calculated by the trapezoidal method between 0 and 120 min. The oral disposition index (DI), which is a composite measure of beta-cell function, was calculated as ΔI_0-30_/ΔG_0-30_ × 1/FSI [μg/mL] [24]. 

The diet of the participants during the previous month was assessed using a validated, brief self-administered diet history questionnaire (BDHQ). At baseline and during the intervention period, trained dietitians completed a 58 item BDHQ during a face-to-face interview with each participant [25].

Physical activity during the previous week was assessed using a validated international physical activity questionnaire (IPAQ) [26]. Energy expenditure as a result of physical activity was calculated using the following formula: physical activity energy expenditure (PAEE) [kcal/day] = physical activity [metabolic equivalents (METs)] × 3.5 [mL/kg/min] × 0.005 [kcal/mL] × weight [kg], where oxygen consumption = 0.005 kcal and 1 METs = 3.5 mL/kg/min) [27]. Body composition was measured using bioelectrical impedance analysis (BIA) with an Inbody 570 (InBody Japan). Height and body mass were measured to the nearest 0.1 cm and 0.1 kg, respectively, by the research dietitians, with participants wearing light clothing and no shoes. BMI was calculated as body weight [kg] divided by the square of body height [m^2^]. Blood pressure (average of two measurements) was measured by the research dietitians or participants themselves using sphygmomanometers. Information on the background and personal behavior of the participants was collected using a questionnaire, comprising eleven questions about family structure, education, smoking habits, medical history, current medication use, and family history of diabetes.

### 2.6. Statistical Analysis

The sample size needed to detect a difference between a pairs of continuous data was estimated from preliminary investigations using G*Power 3.1.7 software [28]. In this way we estimated that with a sample size of 29, the study would have 80% power (1−beta error) to detect an 18% difference in glucose AUC between baseline and the end of the intervention period, with a level of significance of 5%.

The participants were assigned to one of two groups (weighted GRS ≤ 1.10, *n* = 17: L-GRS group and weighted GRS > 1.10, *n* = 15: H-GRS group) using the median weighted GRS for the five T2DM susceptibility genes. Results are presented as number, percentage, mean ± standard deviation (SD), mean ± standard error (SE), or mean difference (95% confidence interval [CI]). Before analysis, data distribution and normality (skewness and kurtosis) were checked. Categorical data and Hardy–Weinberg equilibrium (HWE) were compared using Pearson’s chi-square tests. Associations between PPG and HOMA-IR, and five T2DM susceptibility genes, were estimated using analysis of covariance (ANCOVA). The differences between baseline and the end of the intervention period were analyzed using paired *t*-tests.

All statistical analyses were two-tailed and performed using SPSS for Windows (Version 20 SPSS, Chicago, USA). Statistical analysis of outcome data was conducted on a per-protocol basis. *P* < 0.05 was considered to represent statistical significance.

## 3. Results

An outline of this study is shown in Figure 1. We sent an email to a target population of 467 regarding participation in this study, and 35 healthy participants were included. Among the participants, 32 participants completed the entire study (age range 20–23 years), and three participants (*n* = 3, 9%) dropped out of the study. The overall rate of compliance with the yogurt regimen was 97.4% and there were no adverse events associated with it.

Table 1 shows the allele frequencies of the T2DM susceptibility genes among the participants. There were significant differences in risk allele frequency of KCNQ1 and CDKN2B between the H-GRS and L-GRS groups, but not with regard to CDKAL1, UBE2E2, and IGF2BP2. However, there were no significant differences in the other measured parameters between the two groups. In addition, there were no significant differences in HWE with respect to the five T2DM susceptibility genes in the participants (KCNQ1: *p* = 0.28, CDKAL1: *p* = 0.63, CDKN2B: *p* = 0.46, UBE2E2: *p* = 0.81 and IGF2BP2: *p* = 0.06).

The clinical characteristics of the participants are shown in Table 2. At baseline, glucose AUC_0–30min_ (1338 ± 176 mg/dL.min vs. 944 ± 141 mg/dL.min, *p* = 0.003), glucose AUC_30–120min_ (837 ± 117 mg/dL.min vs. 404 ± 94 mg/dL.min, *p* < 0.001), glucose AUC_0–120min_ (2175 ± 248 mg/dL.min vs. 1348 ± 199 mg/dL.min, *p* < 0.001), TC (196 ± 14 mg/dL vs. 173 ± 13 mg/dL, *p* = 0.005), and LDL-C (113 ± 11 mg/dL vs. 95 ± 10 mg/dL, *p* = 0.006) were higher in the H-GRS than the L-GRS group. Furthermore, insulin AUC_0–30min_ (76 ± 153 μg/mL.min vs. 358 ± 123 μg/mL.min, *p* = 0.003), insulin AUC_0–120min_ (145 ± 316 μg/mL.min vs. 571 ± 253 μg/mL.min, *p* = 0.020), IGI (0.05 ± 0.11 vs. 0.30 ± 0.09, *p* < 0.001), IncAUC_ins/glu_ (0.04 ± 0.09 vs. 0.21 ± 0.07, *p* = 0.002), and DI (0.06 ± 0.27 vs. 0.46 ± 0.21, *p* = 0.006) were lower in the H-GRS than the L-GRS group. 

After the intervention period, glucose AUC_0–30min_ (1116 ± 223 mg/dL.min vs. 1062 ± 187 mg/dL.min, *p* = 0.731), glucose AUC_30–120min_ (600 ± 186 mg/dL.min vs. 452 ± 156 mg/dL.min, *p* = 0.269), glucose AUC_0–120min_ (1717 ± 374 mg/dL.min vs. 1514 ± 313 mg/dL.min, *p* = 0.448), insulin AUC_0–30min_ (68 ± 123 μg/mL.min vs. 228 ± 103 μg/mL.min, *p* = 0.072), IGI (0.06 ± 0.14 vs. 0.22 ± 0.12, *p* = 0.068), IncAUC_ins/glu_ (0.05 ± 0.33 vs. 0.44 ± 0.28, *p* = 0.126), and DI (0.06 ± 0.18 vs. 0.30 ± 0.15, *p* = 0.096) were no different between the groups. However, in H-GRS group, FSI (5.51 ± 1.17 μg/mL to 3.95 ± 1.23 μg/mL, *p* = 0.012), TC (196 ± 14 mg/dL to 185 ± 10 mg/dL, *p* = 0.028), LDL-C (113 ± 11 mg/dL to 105 ± 8 mg/dL, *p* = 0.029), and HOMA-IR (1.14 ± 0.25 to 0.80 ± 0.27 mg/dL, *p* = 0.016) were lower after the intervention than at baseline. In L-GRS group, TC (173 ± 13 mg/dL to 165 ± 10 mg/dL, *p* = 0.032) and TG (70 ± 11 mg/dL to 56 ± 12 mg/dL, *p* < 0.001) were lower after the intervention than at baseline.

The changes in dietary intake, body composition, and physical activity over the four weeks period of yogurt consumption are shown in Table 3. There were no significant differences between the baseline characteristics of the L-GRS and H-GRS groups. Energy intake after the intervention was higher than at baseline in the L-GRS group (1458 ± 178 kcal/day to 1766 ± 193 kcal/day, *p* = 0.003). In addition, body weight (L-GRS; 55.2 ± 2.4 kg to 54.8 ± 2.3 kg, *p* = 0.041, H-GRS; 52.5 ± 2.8 kg to 51.7 ± 2.6 kg, *p* = 0.038) and body fat mass (L-GRS; 14.3 ± 1.5 kg to 12.7 ± 1.5 kg, *p* = 0.022, H-GRS; 13.0 ± 1.7 kg to 12.2 ± 1.7 kg, *p* = 0.022) after the intervention were significantly lower than at baseline in both the L-GRS and H-GRS groups. However, the PAEE during this were not significantly different either between the two groups or between baseline and completion of the study.

## 4. Discussion

In this study, the presence of multiple risk alleles for T2DM susceptibility genes is associated with high PPG and low postprandial serum insulin (PSI). In addition, yogurt intake for four weeks improved PPG, PSI, and HOMA-IR in H-GRS groups (Table 2). To our knowledge, this is the first report to suggest that habitual yogurt consumption ameliorates high PPG and impaired insulin responses in healthy subjects with a high genetic risk of T2DM.

Yamauchi et al. reported that T2DM susceptibility genes identified using GWAS are associated with a 14–41% increase in the risk of diabetes [10]. In the present study, there were no differences in the risk allele frequencies from the previous GWAS. In addition, the five T2DM susceptibility genes were not in HWE in the participants. Therefore, analysis of the five T2DM susceptibility genes and the recruited participants were suitable for the study.

Compliance with the intervention was evaluated by a validated BDHQ that included 58 food and beverage items, and IPAQ. The participants showed high compliance (97.4%) with the yogurt intake for the full study period. Moreover, during the intervention period there were no differences in energy intake or physical activity from baseline in H-GRS group (Table 3). Thus, there were no changes in personal behavior during the study.

We were able to show that higher glucose-AUC and lower DI is associated with a high combined risk allele score. Previously, it has been shown that a low DI is associated with glucose intolerance and is highly predictive of future diabetes [29]. Furthermore, a previous study has demonstrated that the composite measure DI can be used to assess β-cell function, and is independently associated with the risk of developing diabetes [24]. Thus, the DI represents a suitable way of assessing β-cell function, and it could be used to identify subjects with poor β-cell function for interventions. The novelty of our study is the fact that we assessed glucose and insulin tolerance following the ingestion of a test meal, and the use of an extended risk allele score that included five proven β-cell loci [10,30]. We divided the participants into two approximately equally sized groups. ‘t Hart et al. recently reported that individuals with 7–8 risk alleles had 23% lower first phase insulin secretion after a 10 mmol/L glucose clamp than individuals with a lower number of risk alleles [13]. Consistent with this, we have demonstrated that our combined risk allele score was associated with a smaller reduction in insulin AUC, IGI, IncAUC_ins/glu_, and DI (Table 2).

Rideout et al. reported that consumption of low-fat dairy foods for 6 months resulted in lower plasma insulin and insulin resistance, estimated using HOMA-IR, in healthy adults [31]. In addition, Akter et al. demonstrated that the consumption of full-fat dairy products may be associated with lower HOMA-IR in Japanese adults [32]. We have demonstrated that consumption of 150 g yogurt daily for four weeks is associated with lower FSI, PSI, and HOMA-IR in the H-GRS group, but not in the L-GRS group. However, previous studies recruited participants that were obese or overweight, and had the metabolic syndrome [31,33]. We can speculate that a higher risk population might receive a greater benefit from yogurt consumption than a lower risk population.

One potential mechanism underlying the observed metabolic benefits may be yogurt-induced improvements in gut microbial composition. Previous studies have shown an association between gut microbiota and chronic diseases, such as T2DM and obesity [34,35]. Klein et al. reported that daily intake of 300 g yogurt containing probiotic strains improved the fecal microbiota in healthy volunteers [36]. Short-chain fatty acids (SCFAs) such as acetate, propionate, and butyrate, can be used for *de novo* synthesis of lipids and glucose [37], which are the main energy sources for the host. In particular, two orphan G protein-coupled receptors, GPR41 and GPR43, have been reported to be activated by SCFAs [38]. Kimura et al. recently demonstrated that Gpr43-deficient mice are obese even when consuming a normal diet, whereas mice overexpressing this receptor specifically in adipose tissue remain lean, independent of their calorie consumption [5]. They further demonstrated that SCFA-mediated activation of Gpr43 resulted in suppression of insulin signaling in adipose tissue, preventing fat accumulation. In our study, HOMA-IR and body fat mass were reduced by the intervention in the H-GRS group (Table 2 and Table 3). Therefore, changes in gut microbial composition may be responsible for the observed improvements in glucose and insulin responses.

However, some study limitations should be noted. First, our study was a single-arm intervention study, and therefore it will be useful to conduct a randomized crossover study in the future. Furthermore, because only the acute effects of the intervention on glucose and insulin responses were measured, we do not know what the long-term effects of yogurt consumption may be. Second, our study was small, and only a few T2DM susceptibility genes were identified. For these reasons, the risk alleles of several T2DM susceptibility genes tended to be more numerous in the H-GRS group than in the L-GRS group, but there was no significant relationship. With T2DM susceptibility genes having been reported at more than 80 loci worldwide [9], misclassification bias was possible in our study. Third, gut microbial profiling was not performed, nor were biomarkers of yogurt consumption measured, in order to provide an objective measure of compliance with the intervention. However, compliance was reported to be high by self-report, and therefore it is likely that the gut microbiota were affected. Indeed, after the intervention period participants reported higher stool frequency than at baseline (Appendix A). Fourth, to avoid confounding effects, habitual consumers of dairy products were excluded from the study. Therefore, study participants were not randomly selected, but rather were volunteers. It is therefore likely that the participants were relatively health-conscious, and that their dietary habits differed from the general population. Therefore, a well-designed study with a larger sample size that further evaluates many T2DM susceptibility genes is needed.

## 5. Conclusions

A weighted GRS incorporating five validated T2DM susceptibility genes is an independent risk factor for high PPG and low PSI. However, yogurt consumption improved PPG, PSI, and HOMA-IR beyond the effects of standard clinical management. This study suggests that as part of a nutritional therapy, yogurt consumption might be able to reduce the T2DM incidence in the future.

## Figures and Tables

**Figure 1 nutrients-10-01834-f001:**
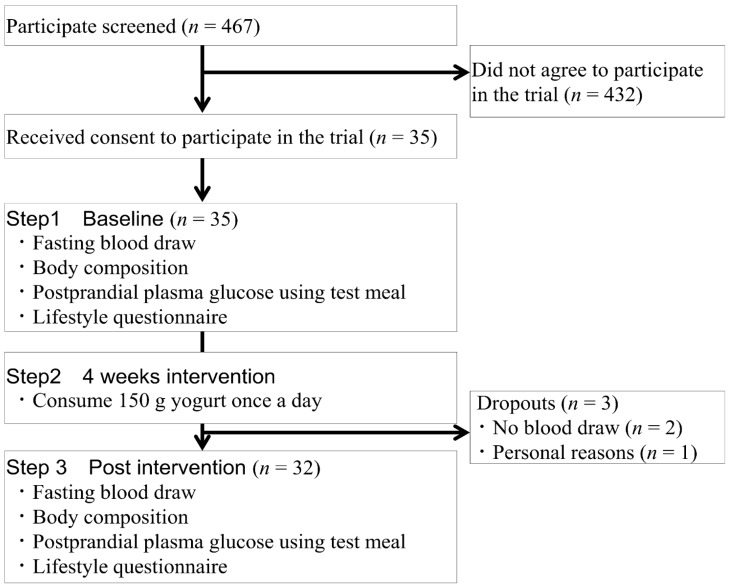
Subject recruitment and study outline. The test meal used 147 g rice, which contained 217 kcal energy, 50 g carbohydrate, and 150 mL water.

**Table 1 nutrients-10-01834-t001:** Subject recruitment and progression throughout the study ^a^.

	L-GRS (*n* = 17)	H-GRS (*n* = 15)	All (*n* = 32)	*p* Value
Age (years) ^b^	20.5 ± 0.8	20.7 ± 1.0	20.3 ± 0.6	0.665
% men (%) ^c^	11.8	6.7	9.4	0.618
Current smoker (%) ^c^	0	0	0	1.000
Height (cm) ^b^	161.8 ± 5.6	159.9 ± 6.5	160.9 ± 6.0	0.382
Weight (kg) ^b^	55.2 ± 6.1	52.5 ± 12.7	54.0 ± 9.8	0.447
BMI (kg/m^2^) ^b,d^	21.1 ± 2.0	20.4 ± 3.9	20.8 ± 3.0	0.534
Skeletal muscle mass (kg) ^b^	40.1 ± 5.5	37.2 ± 7.2	38.8 ± 6.3	0.196
Body fat mass (kg) ^b^	14.3 ± 3.3	13.0 ± 7.0	13.7 ± 5.4	0.507
Waist circumference (cm) ^b^	74.2 ± 4.3	72.6 ± 10.9	73.4 ± 8.1	0.562
Family history of diabetes (%) ^c^	41.2	26.7	34.4	0.386
Weighted genetic risk score ^b^	0.85 ± 0.25	1.31 ± 0.13	1.07 ± 0.20	<0.001
Genetic risk score ^b^	4.6 ± 1.4	7.1 ± 0.8	5.8 ± 1.2	<0.001
KCNQ1 (rs2237892) ^c,e^				
**C**/**C**	2 (11.8)	9 (60.0)	11 (34.4)	0.012
**C**/T	11 (64.7)	5 (33.3)	16 (50.0)	
T/T	4 (23.5)	1 (6.7)	5 (15.6)	
CDKAL1 (rs2206734) ^c,e^				
**T**/**T**	1 (5.9)	5 (33.3)	6 (18.8)	0.096
**T**/C	9 (52.9)	7 (46.7)	16 (50.0)	
C/C	7 (41.2)	3 (20.0)	10 (31.2)	
CDKN2B (rs2383208) ^c,e^				
**A**/**A**	1 (5.9)	9 (60.0)	10 (31.2)	<0.001
**A**/G	12 (70.6)	6 (40.0)	18 (56.3)	
G/G	4 (23.5)	0 (0.0)	4 (12.5)	
UBE2E2 (rs6780569) ^c,e^				
**G**/**G**	11 (64.7)	12 (80.0)	23 (71.9)	0.405
**G**/A	5 (29.4)	3 (20.0)	8 (25.0)	
A/A	1 (5.9)	0 (0.0)	1 (3.1)	
IGF2BP2 (rs1470579) ^c,e^				
**C**/**C**	2 (11.7)	4 (26.7)	6 (18.7)	0.384
**C**/A	7 (41.2)	7 (46.7)	14 (43.8)	
A/A	8 (47.1)	4 (26.6)	12 (37.5)	

Abbreviations: BMI; body mass index, CDKAL1; CDK5 regulatory subunit associated protein 1-like 1, CDKN2B; cyclin-dependent kinase inhibitor 2B, GRS; genetic risk score, IGF2BP2; insulin-like growth factor 2 mRNA binding protein 2, KCNQ1; potassium voltage-gated channel, KQT-like subfamily, member 1, SD; standard deviation, UBE2E2; ubiquitin-conjugating enzyme E2. ^a^ The weighted GRS median cut-off was 1.10 (weighted GRS ≤ 1.10, *n* = 17, 53%: L-GRS group and weighted GRS > 1.10, *n* = 15, 47%: H-GRS group). ^b^ Continuous variables are shown mean ± SD and were analyzed using the unpaired t-tests. ^c^ Categorical variables are shown number (%) or % and were analyzed using the chi-square test. ^d^ BMI was calculated as body mass [kg] divided by the square of body height [m^2^]. ^e^ The reported risk allele is indicated in bold [10]. Hardy-Weinberg equilibrium was not present in the participants (*p* > 0.05).

**Table 2 nutrients-10-01834-t002:** Changes in variables over the four weeks period of yogurt consumption by Japanese adults (*n* = 32).

	Baseline ^a^			Intervention ^a^			Mean Difference ^b^			
	L-GRS	H-GRS	*p* Value	L-GRS	H-GRS	*p* Value	L-GRS	*p* Value	H-GRS	*p* Value
FPG (mg/dL)	80.6 ± 1.8	81.7 ± 2.2	0.565	82.1 ± 2.6	80.0 ± 3.1	0.950	1.5 (−1.0 to 4.0)	0.228	−1.7 (−5.7 to 2.3)	0.388
FSI (μg/mL)	5.43 ± 0.94	5.51 ± 1.17	0.401	4.82 ± 1.03	3.95 ± 1.23	0.484	−0.61 (−2.26 to 1.05)	0.448	−1.55 (−0.40 to −2.71)	**0.012**
Glu-AUC_0–30 min_ (mg/dL.min)	944 ± 141	1338 ± 176	**0.003**	1062 ± 187	1116 ± 223	0.731	118 (−88 to 324)	0.243	−222 (−506 to 63)	0.117
Glu-AUC_30–120 min_ (mg/dL.min)	.404 ± 94	837 ± 117	**<0.001**	452 ± 156	600 ± 186	0.269	48 (−91 to 187)	0.473	−236 (−455 to −18)	**0.036**
Glu-AUC_0–120 min_ (mg/dL.min)	1348 ± 199	2175 ± 248	**<0.001**	1514 ± 313	1717 ± 374	0.448	−166 (−152 to 484)	0.284	−458 (−902 to −15)	**0.044**
Ins-AUC_0–30 min_ (μg/mL.min)	358 ± 123	76 ± 153	**0.003**	228 ± 103	68 ± 123	0.072	−130 (−306 to 45)	0.135	−8 (−76 to 60)	0.802
Ins-AUC_30–120 min_ (μg/mL.min)	213 ± 138	69 ± 172	0.127	222 ± 144	26 ± 172	0.189	9 (−339 to 356)	0.959	−43 (−83 to −3)	**0.039**
Ins-AUC_0–120 min_ (μg/mL.min)	571 ± 253	145 ± 316	**0.020**	450 ± 183	94 ± 218	**0.044**	−122 (−564 to 320)	0.568	−51 (−148 to 46)	0.278
HbA1c (%)	5.1 ± 0.1	5.1 ± 0.1	0.481	5.1 ± 0.1	5.2 ± 0.1	0.192	−0.02 (−0.08 to 0.04)	0.548	0.03 (−0.03 to 0.08)	0.334
TC (mg/dL)	173 ± 13	196 ± 14	**0.005**	165 ± 10	185 ± 10	**0.010**	−8 (−15 to −1)	**0.032**	−11 (−21 to −1)	**0.028**
HDL-C (mg/dL)	65 ± 5	72 ± 5	0.109	64 ± 4	69 ± 4	0.229	−1 (−6 to 3)	0.499	−4 (−8 to 1)	0.086
LDL-C (mg/dL)	95 ± 10	113 ± 11	**0.006**	90 ± 8	105 ± 8	**0.019**	−5 (−10 to 0)	0.055	−8 (−15 to −1)	**0.029**
TG (mg/dL)	70 ± 11	64 ± 11	0.536	56 ± 12	63 ± 11	0.295	−14 (−21 to −7)	**<0.001**	−1 (−17 to 15)	0.903
SBP (mmHg)	98 ± 5	97 ± 6	0.693	97 ± 5	95 ± 5	0.715	−0.4 (−3.6 to 2.9)	0.822	−1.1 (−6.3 to 4.0)	0.644
DBP (mmHg)	54 ± 5	53 ± 5	0.812	55 ± 4	52 ± 4	0.309	1.7 (−1.5 to 4.9)	0.281	−0.4 (−7.5 to 6.7)	0.905
HOMA-IR ^c^	1.09 ± 0.20	1.14 ± 0.25	0.490	0.99 ± 0.22	0.80 ± 0.27	0.495	−0.10 (−0.44 to 0.25)	0.568	−0.34 (−0.61 to −0.07)	**0.016**
HOMA-β ^d^	115 ± 18	104 ± 22	0.162	89 ± 29	63 ± 34	0.962	−27 (−60 to 7)	0.110	−42 (−88 to 5)	0.076
1/FSI	0.23 ± 0.04	0.24 ± 0.06	0.117	0.27 ± 0.09	0.33 ± 0.10	0.471	0.04 (−0.05 to 0.12)	0.368	0.09 (0.15 to 0.03)	**0.012**
Insulinogenic Index ^e^	0.30 ± 0.09	0.05 ± 0.11	**<0.001**	0.22 ± 0.12	0.06 ± 0.14	0.068	−0.07 (−0.22 to 0.07)	0.308	0.01 (−0.04 to 0.06)	0.665
IncAUC_ins/glu_ (μg/mg)	0.21 ± 0.07	0.04 ± 0.09	**0.002**	0.44 ± 0.28	0.05 ± 0.33	0.126	0.24 (−0.29 to 0.76)	0.354	0.02 (−0.02 to 0.06)	0.357
Oral disposition index ^f^	0.46 ± 0.21	0.06 ± 0.27	**0.006**	0.30 ± 0.15	0.06 ± 0.18	0.096	−0.16 (−0.41 to 0.09)	0.190	0.00 (−0.05 to 0.04)	0.866

Abbreviations: ANCOVA; analysis of covariance, CI; confidence interval, DBP; diastolic blood pressure, FPG; fasting plasma glucose, FSI; fasting serum insulin, Glu-AUC; glucose-area under the curve, HbA1c; hemoglobin A1c, HDL-C; high-density lipoprotein cholesterol, HOMA-β; homeostasis model assessment of beta cell function, HOMA-IR; homeostasis model assessment of insulin resistance, Ins-AUC; insulin-area under the curve, IncAUC_ins/glu_; incremental area under the curve insulin/glucose, LDL-C; low-density lipoprotein cholesterol, SBP; systolic blood pressure, SE; standard error, TC; total cholesterol, TG; triglyceride. Bold values are statistically significant (*p* < 0.05). ^a^ The variables are shown mean ± SE and were analyzed using ANCOVA. The FPG, FSI, Glu-AUC, Ins-AUC, HbA1c, HOMA-IR, HOMA-β, 1 / FSI, Insulinogenic Index, IncAUC_ins/glu_, and oral disposition index were adjusted with age, sex, BMI, family history of diabetes, energy intake, physical activity energy expenditure, and TC as covariates. The TC, HDL-C, LDL-C, TG, SBP, and DBP were adjusted with age, sex, BMI, family history of diabetes, energy intake, physical activity energy expenditure, and Glu-AUC_0-120min_ as covariates. ^b^ Mean difference was determined between post-intervention and baseline. The variables are shown mean difference (95% CI) and were analyzed using paired t-test. ^c^ HOMA-IR = FPG [mg/dL] × FSI [μg/L]/(18 × 22.5). ^d^ HOMA-β = 360 × FSI [μg/L]/(FPG [mg/dL] – 63). ^e^ Insulinogenic Index was calculated as the ratio of the change in insulin to the change in glucose between 0 and 30 min (Δ_I 0–30_ / Δ_G 0–30_). ^f^ Oral disposition index was calculated as Δ_I 0–30_/Δ_G 0–30_ × 1/FSI.

**Table 3 nutrients-10-01834-t003:** Changes in dietary intake, body composition, and physical activity over the four weeks period of yogurt consumption by Japanese adults (*n* = 32).

	Baseline ^a^			Intervention ^a^			Mean Difference ^b^			
	L-GRS	H-GRS	*p* Value	L-GRS	H-GRS	*p* Value	L-GRS	*p* Value	H-GRS	*p* Value
Energy (kcal/day)	1458 ± 178	1632 ± 203	0.175	1766 ± 193	1607 ± 220	0.717	308 (89 to 497)	**0.003**	−25 (−240 to 191)	0.810
Protein (g/day)	50.5 ± 6.8	58.3 ± 7.7	0.114	68.7 ± 7.0	63.1 ± 8.0	0.811	18.2 (12.2 to 24.1)	**<0.001**	4.8 (−3.0 to 12.6)	0.208
Fat (g/day)	46.0 ± 5.7	49.7 ± 6.4	0.309	53.6 ± 6.1	49.7 ± 6.9	0.788	7.6 (1.9 to 13.3)	**0.012**	0.0 (−7.9 to 8.0)	0.998
Carbohydrate (g/day)	194 ± 26	221 ± 30	0.176	231 ± 30	214 ± 34	0.833	37 (10 to 65)	**0.011**	−6 (−37 to 24)	0.658
Calcium (mg/day)	302 ± 48	384 ± 55	0.074	782 ± 61	735 ± 69	0.784	480 (377 to 583)	**<0.001**	352 (289 to 415)	**<0.001**
Weight (kg)	55.2 ± 2.4	52.5 ± 2.8	0.524	54.8 ± 2.3	51.7 ± 2.6	0.405	−0.5 (−0.9 to −0.1)	**0.041**	−0.8 (−1.5 to −0.1)	**0.038**
Skeletal muscle mass (kg)	40.1 ± 1.6	37.2 ± 1.8	0.141	39.5 ± 1.4	37.2 ± 1.5	0.273	−0.6 (−2.7 to 1.5)	0.550	0.0 (−0.4 to 0.5)	0.839
Body fat mass (kg)	14.3 ± 1.5	13.0 ± 1.7	0.658	12.7 ± 1.5	12.2 ± 1.7	0.858	−1.6 (−2.9 to −0.3)	**0.022**	−0.9 (−1.6 to −0.1)	**0.022**
Waist circumference (cm)	74.2 ± 2.1	72.6 ± 2.4	0.715	73.7 ± 2.1	71.7 ± 2.4	0.612	−0.6 (−1.4 to 0.2)	0.151	−0.8 (−1.8 to 0.1)	0.079
PAEE (kcal/day) ^c^	421 ± 108	328 ± 123	0.789	438 ± 165	298 ± 188	0.546	.17 (−237 to 271)	0.891	−30 (−117 to 58)	0.478

Abbreviations: ANCOVA; analysis of covariance, CI; confidence interval, PAEE; physical activity energy expenditure, SE; standard error. Bold values are statistically significant (*p* < 0.05). ^a^ The variables are shown mean ± SE and were analyzed using ANCOVA, with age, sex, and family history of diabetes as covariates. ^b^ Mean difference was determined between post-intervention and baseline. The variables are shown mean difference (95% CI) and were analyzed using paired t-test. ^c^ PAEE [kcal/day] was calculated as physical activity [Mets.mins] × 3.5 [mL/kg/min] × 0.005 [kcal/mL] × weight [kg], where oxygen consumption = 0.005 kcal and 1 METs = 3.5 mL/kg/min.

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
