# Peer review of "Daily Yogurt Consumption Improves Glucose Metabolism and Insulin Sensitivity in Young Nondiabetic Japanese Subjects with Type-2 Diabetes Risk Alleles"

_nutrients, 2018, doi:10.3390/nu10121834_

Reviewer 1 Report

This is a useful study that adds knowledge to current literature about diabetes risk and dietary intervention, in particularly in Japanese population.

There are obvious flaws in this study, however, the authors are well aware of this and acknowledged this in the discussion.

There are still a few places need improvement.

line 27-8, the conclusive sentence of the abstract needs to be revised to better summarise the indication of the study.

Table 2, the arrangement of numbers and text is not appropriate. Should not separate one number into two lines (i.e. the P value). very hard to reader to understand clearly.

Table 3 has the same problem as Table 2.

Line 281, please change "diabetic complications" into "T2DM incidence", as diabetic complications mainly refer to diabetic eye, kidney, foot, heart complications etc. 

Reviewer 2 Report

1-The sample size (N=35) is insufficient in the context of polygenic association studies.

2-A randomized clinical trial is more appropriate than a single arm intervention study to investigate the interaction between genetic variants, diet intervention and metabolic outcomes.

3-Five loci have been included in this study. However, > 30 loci have been associated with T2D in East Asian populations to date. This means that the genetic risk score is not up to date.

4-The classification of participants into a low versus high genetic risk does not make sense.

Reviewer 3 Report

Authors describe a short interventional study with Young healthy volunteers taking 150 g of non-fatty yogurt per day. The association between yogurt intake and glucose tolerance and pancreatic beta cell function is analyzed according to a genetic risk classification for type 2 diabetes. The study is well designed and described. The study has some limitations that are properly declared in the discussion. But some improvements can be applied:

Participants are classified according to their genetic risk for diabetes using a non-weighted GRS. Only 2 of the 5 selected SNPs contribute mainly to the classification. In the other SNPs, the risk allele is more frequent in the L-GRS group, suggesting a probable misclassification of some participants. I suggest to weighted the GRS according to the OR for diabetes reported for those SNPs in the Asian population and re-classify to the individuals according to the median of the GRS or in tertiles (although the sample size is limited).

There is a significant increase in energy intake in the L-GRS group involving also an increase in carbohydrate and fats and a higher decrease in body mass and fat mass in comparison with the H-GRS group. Authors have not taken these changes into account in analyses but they could be affecting glucose and diabetes determinations. Authors should re-analyze including these confusion factors.

Author Response

Round  2

Reviewer 3 Report

Authors properly incorporated my suggestoins

Author Response

Responses to the comments of Reviewer 3

We thank the Reviewer for the constructive comments regarding our paper. We have made new changes to the paper that reflect the new comments. Additionally, we had the revised manuscript checked by a professional medical manuscript editor who is a native speaker of English. In previous review comments, we thank very important comment (weighted Genetic Risk Score [GRS]). The results from weighted GRS had more clear results than non- weighed GRS on glucose and insulin responses. We would like to express my gratitude to you once again.
